# Reliable Service Function Chain Deployment Method Based on Deep Reinforcement Learning

**DOI:** 10.3390/s21082733

**Published:** 2021-04-13

**Authors:** Hua Qu, Ke Wang, Jihong Zhao

**Affiliations:** 1School of Software Engineering, Xi’an Jiaotong University, Xi’an 710049, China; qh@mail.xjtu.edu.cn; 2School of Electronic and Information Engineering, Xi’an Jiaotong University, Xi’an 710049, China; zhaojihong@xjtu.edu.cn; 3School of Communication and Information Engineering, Xi’an University of Posts & Telecommunications, Xi’an 710061, China

**Keywords:** network function virtualization, service function chain, reliable, priority-awareness, deep reinforcement learning

## Abstract

Network function virtualization (NFV) is a key technology to decouple hardware device and software function. Several virtual network functions (VNFs) combine into a function sequence in a certain order, that is defined as service function chain (SFC). A significant challenge is guaranteeing reliability. First, deployment server is selected to place VNF, then, backup server is determined to place the VNF as a backup which is running when deployment server is failed. Moreover, how to determine the accurate locations dynamically with machine learning is challenging. This paper focuses on resource requirements of SFC to measure its priority meanwhile calculates node priority by current resource capacity and node degree, then, a novel priority-awareness deep reinforcement learning (PA-DRL) algorithm is proposed to implement reliable SFC dynamically. PA-DRL determines the backup scheme of each VNF, then, the model jointly utilizes delay, load balancing of network as feedback factors to optimize the quality of service. In the experimental results, resource efficient utilization, survival rate, and load balancing of PA-DRL were improved by 36.7%, 35.1%, and 78.9% on average compared with benchmark algorithm respectively, average delay was reduced by 14.9%. Therefore, PA-DRL can effectively improve reliability and optimization targets compared with other benchmark methods.

## 1. Introduction

In the traditional network, a plenty of network functions are relying on dedicated hardware deployment to form a variety of network services, this traditional deployment method can no longer meet the complex and changeable network requirements with the rapid growth of the number and diversity of network service requests. However, the emergence of network function virtualization (NFV) technology has solved that problem [1]. NFV makes network equipment functions no longer depend on dedicated equipment through software and hardware decoupling and function abstraction, in addition, NFV can reduce the high cost of network equipment by sharing network resources fully and flexibly. Moreover, NFV can realize the rapid development and deployment of new business, and realize automatic deployment, elastic scaling, fault isolation and self-healing of new business, etc. [2]. In NFV, several virtualized network functions (VNF) form a sequence of network functions that can handle specific network services by deploying on server nodes in a certain order, which is named as service function chain (SFC) [3]. As an important form of network and service, the SFC can construct a complete end-to-end network service through rational orchestration of VNFs [4].

Reliable deployment is an important concern for many SFCs, the traditional SFC deployment method relies on the routing strategy formulated by service providers, that makes network traffic go through several VNFs in a certain order to provide the required network functions [5]. However, there are some defects need be solved. First, the method is lack of certain tolerance measure which is quite important for SFC deployment, so that it’s difficult to meet the reliable deployment requirements of SFC. Secondly, different SFC’s priority is not the same because of difference in resource requirements, so simple strategy is hard to meet all deployment requirements. Last but not least, it’s difficult to response active and complex SFC deployment requirements because VNF is pre-deployed on server node in traditional network, so that the VNFs are not flexible to satisfy the SFCs.

In view of the dynamic and reliable deployment of the service function chain, the existing deployment methods have some shortcomings. Previously, most SFC deployment models only solve the problem which is determining deployment location or achieving optimization targets [6,7,8,9]. However, the reliability requirements have increased with the development of NFV and increasing of SFC requests. To this end, reliable and fault-tolerant SFC deployment methods are proposed to achieve high reliability [10,11,12,13,14,15]. Specifically, reliable SFC deployment methods usually use backup scheme to deploy more VNF instances on different network nodes. However, traditional backup scheme is too simple to satisfy complex SFC requests. On the other hand, more advanced and powerful reinforcement learning models have significantly achieved performance gains in SFC deployment problem recently [16,17,18]. However, Q-learning, a classical algorithm of reinforcement learning, needs to maintain a quite large Q-table because of large state and action set, so that the computing power of the algorithm will be affected with wasting of CPU and memory. To this end, the deep reinforcement learning which is more advanced than reinforcement learning is proposed to apply on SFC deployment problem [19,20,21,22]. Specifically, the deep reinforcement learning can choose better deployment and backup locations with learning process, and it can make accurate judgements about unknown situations in the sample.

To solve the dynamic and reliable SFC deployment problem, this paper formulates the reliable SFC deployment framework and proposes a novel priority awareness deep reinforcement learning (PA-DRL) algorithm to deploy and backup VNFs with high reliability and optimization performance. The main contributions of this paper are as follows.

Firstly, we propose the priority awareness framework to determine the priority of SFC and network node. On one hand, we calculate the SFC priority by average proportion of CPU resource and bandwidth resource requirements. On the other hand, we calculate the network node priority by average proportion of node resources and node degrees. It is the first time that determining the overall priority through SFC and node priority in our work.Secondly, we design a determining scheme of backup location set for VNF. The determining scheme can predetermine the backup location set by SFC and node priority which are mentioned above. Specifically, the scheme can decide whether to backup and guarantee that deployment and backup locations are different of each VNF.Lastly, we apply deep reinforcement learning algorithm on the process of VNF deployment and backup, then we propose PA-DRL model to deploy and backup VNFs with reliability requirement. Specifically, we design the feedback function with three factors including transmission delay, node balancing, and link balancing. Then, we evaluate the proposed PA-DRL model on the network topology which is randomly generated, and the extensive experimental results demonstrate that PA-DRL can effectively deploy SFC requests and achieve deployment performance compared with other four models including two widely used SFC deployment methods and other two backup schemes of PA-DRL.

The rest of this paper is organized as follows. Section 2 surveys the related works. In Section 3, the priority awareness framework used for determining backup scheme is described and then the proposed PA-DRL algorithm to solve reliable SFC deployment is discussed in details. In Section 4, the experimental designs and simulation results are shown. Finally, the conclusion and discussion of future work are given in Section 5.

## 2. Related Works

In the early studies, SFC deployment problem is usually modeled as an optimization problem, which is most solved by integer linear programming (ILP) and heuristic algorithm. On the one hand, a part of the author uses ILP algorithm or its deformation to solve the SFC deployment optimization problem. For example, Insun Jang et al. proposed a polynomial time algorithm based on linear relaxation and rounding to approximate the optimal solution of the mixed ILP [6]. Although they increased SFC acceptance rate and the service capacity, yet more optimization targets are needed, including transmission delay, load balancing and so on. Zhong et al. orchestrated SFCs across multiple data centers and proposed an ILP model to minimize the total cost [7]. They reduced the overall cost, however, the optimization result is too simple to apply in most cases. Although the ILP as a statical algorithm can solve most SFC deployment optimization problem, yet it is difficult to apply on dynamic and complex SFC requests. On the other hand, other authors solve the optimization problem by heuristic algorithm. For example, Li et al. described the SFC deployment as a multi-objective and multi-restriction problem and proposed a heuristic service function chain deployment algorithm based on longest function assignment sequence [8]. Although they jointly optimized the number of VNF deployment and link bandwidth requirements, yet they ignored the dynamicity of the network. Mohammad Ali Khoshkholghi et al. formulated a multi-objective optimization model to joint VNF placement and link embedding in order to reduce deployment cost and service latency, then proposed two heuristic-based algorithms that perform close to optimal for large scale cloud/edge environments to solve the optimization problem [9]. They optimized both cost and delay, however, the method was needed to improve to apply in dynamic network environment. Heuristic algorithm can solve the SFC deployment problem which is NP-hard effectively, however, the heuristic algorithm is easy to fall into the local optimum, and the parameter setting is affected by the experience value, the number of algorithm iterations and convergence speed are difficult to guarantee.

More recently, reinforcement learning and deep reinforcement learning models have shown their superior capabilities in SFC deployment due to their advanced feature extraction and data modeling abilities. For example, Wei et al. designed a service chain mapping algorithm based on reinforcement learning to reduce the average transmission delay and improve the load balancing, the algorithm determined location of each VNF according to the network status and reward value of feedback function after the deployment [16], however, they did not improve the reliability of SFC which is significant in current network. Sang Il Kim et al. considered the consumption of CPU and memory resources, then utilized reinforcement learning to solve SFC optimization problem dynamically [17], yet the algorithm was not satisfied with delay sensitive SFC. Sebastian Troia et al. investigated the application of reinforcement learning for performing dynamic SFC resources allocation in NFV-SDN enabled metro-core optical networks [18], they decided how to reconfigure the SFCs according to state of the network and historical traffic traces. However, reinforcement learning exists the limitation that it cannot satisfy the networks which has large-scale states. To this end, deep reinforcement learning is proposed to solve the large-scale SFC deployment problem. For example, Fu et al. decomposed the complex VNFs into smaller VNF components to make more effective decisions and proposed a deep reinforcement learning-based scheme with experience replay [19], although they solved dynamic and complex SFC deployment problem effectively, yet more optimization targets and reliability needed to be considered. Li et al. proposed an adaptive deep Q-learning based SFC mapping approach to improve CPU and bandwidth resource utilization rates [20], they improved whole system resource efficiency, however, they ignored the transmission delay in SFC deployment problem. Pei et al. used deep reinforcement learning to improve network performance, including SFC acceptance rate, throughput, end-to-end delay, and load balancing [21], the method optimized a variety of objectives, however, the SFC reliability was not considered in the optimization process, which may lead to SFC failure. Although these researches have solved the large-scale SFC deployment problem effectively, yet they ignored the reliability of SFC, and cannot satisfy to recovery SFC after network failure.

For reliability and fault-tolerance, several reliable SFC deployment models are proposed to achieve high reliability. For example, Francisco Carpio and Admela Jukan improved service reliability using jointly replications and migrations, and then proposed a N2N algorithm based on LP to improve reliability and network load balancing [10]. However, N2N cannot solve dynamic SFC requests effectively, and it cannot guarantee performance of transmission delay in results. Abdelhamid Alleg et al. proposed a reliable placement solution of SFC modeled as a mixed ILP program, they designed the SFC availability level as a target and reduce the inherent cost which is affected by diversity and redundancy [11]. However, the proposed model is only applied on statical placement and lack of adaptability of machine learning techniques, on the other hand, the algorithm does not distinguish SFC priority, so that all SFC requests applied the same scheme which would lead to low effective utilization of CPU resources. Tuan-Minh Pham et al. designed the UNIT restoration model and the PAR protection algorithm based on ILP framework, the proposed algorithms protect SFC requests from network failures in terms of both resource restoration and recovery time efficiently [12]. Although the model achieves robustness of network, yet it does not guarantee the delay in a service demand, moreover, the solution needs to consider the dynamic parameters of online demands. Mohammad Karimzadeh-Farshbafan et al. proposed a polynomial time sub-optimal algorithm named VRSP which is based on mixed ILP in multi-infrastructure network provider environment [13]. However, VRSP solves the reliable SFC deployment problem only by statical method, moreover, VRSP does not consider transmission delay and network balancing in the optimization process. Qu et al. formulated the reliable SFC deployment problem as a mixed ILP and proposed a delay-aware hybrid shortest path-based heuristic algorithm [14]. Although they achieved high reliability and low latency through the model based on ILP and heuristic algorithm, yet the model is too simple to satisfy complex SFC deployment because it does not distinguish the priority of SFC, and the algorithm cannot predict deploy and backup scheme of unknown SFC requests accurately. Ye et al. proposed a novel heuristic algorithm to efficiently address the joint topology design and mapping of reliable SFC deployment problem [15]. However, the model does not consider the delay and load balancing as optimization targets, and heuristic algorithm is easy to fall into the local optimum unlike machine learning algorithm such as reinforcement learning and deep reinforcement learning. Mao et al. proposed a deep reinforcement learning based online SFC placement method to guarantee seamless redirection after failures and ensure service reliability [22]. Although the model automatically deploys both active and standby instances in real-time, yet it does not distinguish the priority of SFC and cannot guarantee transmission delay and network load balancing. Motivated by it, this paper will for the first-time design priority-awareness model, then applies deep reinforcement learning to solve reliable SFC deployment problem in NFV framework.

## 3. The Proposed PA-DRL Framework

In this section, we first formulate the reliable SFC deployment problem. Next, we discuss the priority awareness framework of SFC and network node in details, and then describe the rules for determining backup scheme of each VNF. Finally, the proposed deployment and backup algorithm based on deep reinforcement learning is explained.

### 3.1. Reliable SFC Deployment Formulation

The reliability of SFC deployment is aimed to find a suitable backup scheme for each VNF. In this work, we define the physical network topology as an undirected graph, which is represented as G=(V,E), where V is the set of network nodes and $\left|V\right|=n_{V}$, E denotes the set of links that connect two network nodes directly and $\left|E\right|=n_{E}$. In addition, we define the set of SFC requests as $Req=\left\{sfc_{1},sfc_{2},\ldots ,sfc_{n}\right\}$, and each SFC request consists of several different VNFs and virtual links between two adjacent VNFs.

Because network hosts VNF and virtual link via CPU and bandwidth resources respectively, therefore we mainly consider CPU resources on network nodes and bandwidth resources on physical links. Due to the difference of computing power requirements, we assume that the number of CPUs occupied by different VNF is not same. Similarly, we assume that the amount of bandwidth resources occupied by different virtual link is different too. Moreover, in order to quantify the network load balancing, we define variance of CPU’ percentage occupied on all network nodes as an indicator,
(1)$$bal_{n}=\overset{n_{V}}{\underset{i=1}{\sum }}{\left(\frac{cpu_{i}}{CPU_{i}}-\overset{n_{V}}{\underset{j=1}{\sum }}\frac{cpu_{j}}{CPU_{j}}/n_{V}\right)}^{2}/n_{V},$$
where $cpu_{i}$ and $CPU_{i}$ indicate remaining and total vCPU resources on node i respectively. And then, we define variance of bandwidth’ percentage occupied on all physical links as an indicator too,
(2)$$bal_{l}=\overset{n_{E}}{\underset{i=1}{\sum }}{\left(\frac{bw_{i}}{BW_{i}}-\overset{n_{E}}{\underset{j=1}{\sum }}\frac{bw_{j}}{BW_{j}}/n_{E}\right)}^{2}/n_{E},$$
where $bw_{i}$ and $BW_{i}$ indicate remaining and total bandwidth resources on link i respectively. In addition, SFCs consist of several different VNFs, so the length of SFCs may has a difference, as a result the delay of different SFCs will be quite different. For that reason, we use average delay of adjacent VNFs to measure the performance of delay optimization,
(3)$$del=\overset{n}{\underset{i=1}{\sum }}\overset{l_{i}}{\underset{j=2}{\sum }}d\left(f_{j-1},f_{j}\right)/\overset{n}{\underset{i=1}{\sum }}\left(l_{i}-1\right),$$
where $l_{i}$ denotes the length of $SFC_{i}$, and $d\left(f_{j-1},f_{j}\right)$ represents the delay between VNF $f_{j-1}$ and $f_{j}$. Besides, there are limits of the network resources, it means that the resource on each node and link is non-negative and no more than the maximum resource limit.
(4)$$\{\begin{array}{c} 0\leq cpu_{n}\leq CPU_{n},\;\;\forall n\in V\\ 0\leq bw_{l}\leq BW_{l},\;\;\forall l\in E \end{array}.$$

### 3.2. Priority Awareness of SFC and Network Node

Priority awareness is aimed to determine the important level of SFC and node through resource requirements and network status. Firstly, the algorithm determines the priority of SFC in high, medium, and low levels. Then, the algorithm judges the priority of nodes according to the network status. Finally, the model decides whether to allocate CPU resources to each VNF’s backup. The algorithm determines the backup scheme of each VNF through these steps, then, the deployment algorithm will deploy each VNF and its backup.

In this work, we define effective resource utilization as the proportion of deployment resources and backup resources, which is expressed as,
(5)$$e=\overset{n}{\underset{i=1}{\sum }}\overset{l_{i}}{\underset{j=1}{\sum }}c_{d}/\left(\overset{n}{\underset{i=1}{\sum }}\overset{l_{i}}{\underset{j=1}{\sum }}c_{d}+\overset{n}{\underset{i=1}{\sum }}\overset{l_{i}}{\underset{j=1}{\sum }}c_{b}\right),$$
where $c_{d}$ and $c_{b}$ denote CPU resources which are occupied by VNF’s deployment and backup respectively. To improve the effective utilization of resource on network nodes, we propose the process of priority awareness, which aims to choose a suitable backup scheme for each VNF.

At first, we use the number of required resources to measure a SFC’s priority, which is defined as,
(6)$$p_{sfc}=\frac{\sum_{i=1}^{l}rc_{i}/m_{rc}+\sum_{i=2}^{l}rb\left(f_{i-1},f_{i}\right)/m_{rb}}{2},$$
where $rc_{i}$ denotes the required CPU resources of VNF $f_{i}$, $rb\left(f_{i-1},f_{i}\right)$ denotes the required bandwidth resources of virtual link between $f_{i-1}$ and $f_{i}$, $m_{rc}$ and $m_{rb}$ denote the maximum required CPU and bandwidth resources of all SFCs respectively.

Secondly, we use the resource state and node degree to measure a node’s priority, which is defined as,
(7)$$p_{node}=\frac{(CPU_{i}-cpu_{i})/CPU_{i}+d_{i}/\left(\frac{\sum_{i=1}^{n_{V}}d_{i}}{2}\right)}{2},$$
where $d_{i}$ denotes the node degree of node i.

Finally, we divide SFC into three priority levels, including high, medium, and low respectively. Specifically, the SFC has high priority level if $p_{sfc}>1/3$, then the SFC has low priority level if $p_{sfc}<1/3$, otherwise, the SFC has medium priority level. Further, we divide network node into two priority levels, including high and low respectively. Specifically, the node has high priority level if $p_{node}>0.5$, otherwise its priority level is low.

### 3.3. Rules for Determining Backup Scheme

The backup scheme is choosing whether to backup for each VNF in our work, we define $\left(V_{d},V_{b}\right)$ as the node pair of deployment and backup node for each VNF, so that $V_{b}=0$ if the VNF had no backup, otherwise $V_{b}\neq 0$, besides, there is a constraint $V_{d}\neq V_{b}$ to guarantee that each VNF’s deployment and backup nodes are different. Table 1 presents the backup scheme in different situations, including three SFC’s priority levels and two node’s priority levels as mentioned above. Specifically, when SFC priority is high or SFC priority is medium and node priority is high, VNF is backed up, otherwise, when SFC priority is low or SFC priority is medium and node priority is low, VNF is not backed up.

### 3.4. Deep Reinforcement Learning Deployment Algorithm

Deep reinforcement learning algorithm aims to solve the problem which has large-scale state and action set. In this work, we define the state set as,
(8)$$S\left(t\right)=\left(cpu_{t},bw_{t},vnf_{t}\right),$$
where $cpu_{t}$ and $bw_{t}$ denote the remaining resources of all nodes and links at time t respectively, and $vnf_{t}$ denotes the related information of VNF at time t, including current deployment location, species of VNF at time t, and the species of next VNF at time t+1. We define action set as the node pair $\left(V_{d},V_{b}\right)$ as mentioned before, which is represented as,
(9)$$A\left(t\right)=\left(V_{d},V_{b}\right),$$
so, the number of all actions is $n_{V}{}^{2}$.

We aim to optimize the transmission delay of SFC and network load balancing, including load balancing of nodes and links. To this end, we optimize all optimization targets by minimizing the maximum value of delay, node, and link load balancing, so we define the feedback function as,
(10)$$\{\begin{array}{c} M_{D}=\max\left(d\left(f_{i},f_{i+1}\right)\right)\\ M_{C}=\max\left(CPU_{i}-cpu_{i}\right)\\ M_{B}=\max\left(BW_{i}-bw_{i}\right)\\ R\left(t\right)=\frac{\alpha }{M_{D}}+\frac{\beta }{M_{C}}+\frac{\gamma }{M_{B}} \end{array},$$
where α,β,γ are the weight coefficients, and need to meet this constraint,
(11)$$\{\begin{array}{c} \alpha >0,\beta >0,\gamma >0\\ \alpha +\beta +\gamma =1 \end{array}.$$

At first, we initialize all the parameter ω of neural network. Then we choose action through ε−greedy method, which is defined as,
(12)$$\varepsilon =\frac{K}{100},$$
where K is the exploration times. Finally, we store a 5-tuple $\left(S\left(t\right),A\left(t\right),S\left(t+1\right),R\left(t\right),is_{end}\right)$ in replay memory buffer D after every action selection, where $is_{end}$ is a binary indicates whether the state is terminated,
(13)$$\{\begin{array}{c} is_{end}=1,\;\;\;state\;is\;terminated\\ is_{end}=0,\;\;\;state\;is\;not\;terminated \end{array}.$$

The network randomly selects several samples from D in each episode, then the Q value will be calculated by,
(14)$$Q\left(t\right)=\{\begin{array}{c} R\left(t\right),\;while\;is_{end}=1\\ R\left(t\right)+\delta *\max\left(Q\left(S\left(t+1\right),A\left(t+1\right),\omega \right)\right)\;\\ ,\;while\;is_{end}=0 \end{array},$$
where ω is the parameter of neural network, which will be updated in each episode by backward gradient propagation, and δ represents the learning rate. The training algorithm for our model is as Algorithm 1.
**Algorithm 1** PA-DRL training algorithm**Input:** SFC request with the requirement of VNFs and virtual links**Output:** Deployment and backup locations for each VNF of SFCInitialize the neural network N with the parameter ω, and clear the replay memory buffer D.**while**$is_{end}\neq 0$**do**Calculate SFC priority level $p_{sfc}$ as (6)Calculate node priority level $p_{node}$ as (7)Determine action set by Table 1Select action A(t) by ε−greedy methodCalculate reward R(t) as (10)Store the sample $\left(S\left(t\right),A\left(t\right),S\left(t+1\right),R\left(t\right),is_{end}\right)$ in memory DRandomly select samples from D then calculate Q as (14)Train the network N and update ω by using back propagation of the loss function**end while**

## 4. Simulation Results and Analysis

In this section, the network topology is introduced firstly, and then the parameters are described, and finally the optimization performance and analysis are presented and discussed.

### 4.1. Network Topology

To formulate the deployment process of SFC, we randomly generate a network topology, which has 6 network nodes and 7 physical links, the network topology diagram is showed in Figure 1. The black number represents CPU capacity of each network node, the red and blue numbers next by solid line represent bandwidth capacity and transmission delay of each physical link respectively. The CPU resources of nodes and bandwidth resources of links are set by random method, and we assume all link delay in order to facilitate the measurement of transmission delay.

### 4.2. Experimental Settings

#### 4.2.1. Formulation Environment

The experiments are implemented with MATLAB platform on a 64-bit computer with Intel Core CPU i5-9400F @ 2.9 GHz, 16-GB RAM. In order to avoid linear results due to linear SFC requests number, we randomly generate 4000, 6000, 8000, 10,000, 15,000, 20,000, 25,000, 30,000, 40,000, 50,000 SFC requests respectively.

#### 4.2.2. Comparison Algorithms

In order to measure the performance of PA-DRL algorithm which is proposed in this work, two state-of-the-art methods including PA-RL, N2N are selected as the baselines:

PA-RL, we apply priority-awareness model to RL proposed in [16].

N2N, which is designed based on LP algorithm in [10].

In addition, we implement three version of our solution:

ALL-BACKUP, backs up all VNFs of each SFC no matter what their priority level is.

RND-BACKUP, randomly determines VNFs of each SFC to back up.

PA-DRL, chooses VNFs of each SFC to back up through priority awareness framework as mentioned before.

#### 4.2.3. Related Parameters

VNFs and virtual links. We assume there are five varieties of VNFs, including Firewall (FW), Net Address Transition (NAT), Deep Packet Inspection (DPI), Domain Name System (DNS), and Load Balancing (LB). The required resources of each VNFs and virtual links are presented as Table 2 and Table 3. For example, the SFC FW-NAT-DPI has 4 units CPU and 3 units bandwidth resources requirement respectively. So that, the SFC NAT-DPI-DNS-LB-FW has most resources requirement, the required CPU and bandwidth resources of this SFC are 10 units CPUs and 10 units bandwidths respectively.

#### 4.2.4. Parameters of Algorithm

On one hand, through several times of adjusting according to the experimental performance, we set the parameters of feedback function (10) as α=0.1, β=0.45, and γ=0.45 respectively, and then, we set up two hidden layers with 100 neurons for the neural network. On the other hand, in the training process of algorithm, the learning rate of deep reinforcement learning is usually set as [0.9, 0.999], so that the learning effect will be better than other value the learning rate is. In this work, we set the learning rate δ of proposed deep reinforcement learning process as δ=0.99. Moreover, we set the size of replay memory buffer D is 8000, and the number of samples for each training episode is 1000, so as to ensure that the correlation between samples is small and the distribution of samples is uniform. In addition, to guarantee that there are enough resources to deploy VNFs and virtual links, we assume the life cycle of SFC is the time of 20 SFCs’ living time, which means that the system need to release CPU and bandwidth resources of $sfc_{i-20}$ when $sfc_{i}$ comes (i>20).

### 4.3. Experimental Results

We get the results of runtime, effective utilization of CPU resource, average survival rate of SFC, average delay of adjacent VNFs, nodes’ load balancing variance, and links’ load balancing variance under each SFC requests number as mentioned before.

#### 4.3.1. Runtime

Figure 2 shows the runtime for PA-DRL and PA-RL models in terms of deploy SFCs number. It can be easily observed that the runtime of PA-DRL is much lower than PA-RL under each SFC requests number, and both curves show an increasing trend. Because the PA-RL need to maintain a large-scale Q-table in the process of learning, and need to traverse the whole Q-table when every time that an action is selected, so the time consumption is quite large. However, the PA-DRL proposed in this work uses the deep reinforcement learning framework, eliminates the curse of dimensionality in PA-RL, so that it can save a lot of time when selecting an action. On the other hand, the two methods need to complete more times of selecting action when SFC requests number is getting bigger, so the runtimes of proposed PA-DRL and PA-RL algorithms grow linearly.

#### 4.3.2. Resources’ Efficient Utilization

Figure 3 shows the resources’ efficient utilization of proposed PA-DRL method and other four baseline algorithms. It is easily to observe that the performance of PA-DRL is better than other algorithms. Intuitively, there are three facts in Figure 3. Firstly, the efficient utilization of PA-DRL is the best, because PA-DRL determines the priority of nodes by node resource utilization and node degree, this process can avoid the circumstance that the VNF which is not important on the medium priority SFC occupies CPU resources, so that PA-DRL can avoid wasting of resources and improve efficient utilization of CPU resource. Secondly, the efficient utilizations of RND-BACKUP and PA-RL are better than N2N, because they utilize deep reinforcement learning and reinforcement learning algorithm respectively, which can improve the performance through learning process, yet N2N is based on LP algorithm, which is weaker than machine learning on optimization performance. Last but not least, the efficient utilization performance of ALL-BACKUP method is the worst, because ALL-BACKUP backs up all VNFs no matter which the priority level is, thus, the resources’ efficient utilization is keeping at 50% under each SFC requests number.

#### 4.3.3. Average Survival Rate

Figure 4 shows the average survival rate, which is defined as average SFC survival rate when each node fails, for PA-DRL, ALL-BACKUP, RND-BACKUP, PA-RL, and N2N models in terms of SFCs number, the average survival rate in this work is defined as the average proportion of survival SFCs and all SFC requests when each network node fails. The Figure 4 contributes four facts. At first, it can be easily observed that average survival rate of ALL-BACKUP method is the best, because each VNF in ALL-BACKUP method has one backup on other one node, so that every SFC can be recovered when each network node fails, and the average survival rate is keeping at 100% under all SFC requests. Then, the performance of PA-DRL is better than other algorithms except ALL-BACKUP, because the PA-DRL determines the backup scheme of each VNF by priority of SFC and network node, so that PA-DRL can improve the destruction resistance of high priority SFCs, furthermore, PA-DRL can avoid the circumstance of large-scale SFC failure caused by physical network malfunction. Thirdly, the average survival rate of PA-RL is lower than PA-DRL, because the reinforcement learning algorithm cannot make accurate judgements about unknown situations in the sample compared with deep reinforcement learning. Last but not least, the performances of RND-BACKUP and N2N models are the worst, because their backup schemes are worse than PA-DRL, and PA-DRL’s deep reinforcement learning algorithm is more advanced than N2N.

#### 4.3.4. Average Delay

Figure 5 shows the average delay of proposed PA-DRL algorithm and four compared algorithms. From Figure 5, it is easily to observe three facts. Firstly, the average delay of N2N is the largest, this is because N2N is based on LP algorithm, this algorithm is satisfied for statical environment and cannot improve the optimization performance gradually like deep reinforcement learning and reinforcement learning. Therefore, the performance of N2N is the worst under each SFC requests number. Secondly, this result agrees with the theoretical analysis above that the proposed PA-DRL algorithm is better than PA-RL, because we use deep reinforcement learning algorithm which is more advanced than reinforcement learning, and define maximum delay of adjacent VNFs as one feedback factor of feedback function. Thirdly, the average delay of ALL-BACKUP is less than RND-BACKUP algorithm, because ALL-BACKUP algorithm will back up all VNFs and consume more network resources, the algorithm is more inclined to deploy the VNFs of the same SFC on close nodes, and deploy the backup VNFs on other nodes, so as to balance the network load and reduce the transmission delay.

#### 4.3.5. Load Balancing of Network Nodes

Figure 6 shows the network node load variance for PA-DRL, ALL-BACKUP, RND-BACKUP, RL, and N2N models in terms of SFCs number. Firstly, it can be easily observed that the performance of PA-DRL is the best, because we use maximum CPU usage as optimization goal in feedback function, the performance will be the best through learning process. Secondly, the variance of RND-BACKUP and ALL-BACKUP models are higher than PA-DRL and lower than PA-RL. On one hand, because they utilize maximum node load to optimize the load balancing of nodes, and then, deep reinforcement learning algorithm is more advanced than reinforcement learning in choosing action for unknown samples. On the other hand, because they determine the backup scheme without priority level, the backup scheme of each VNF is worse than PA-DRL, thus, their performances of nodes’ load balancing are worse than PA-DRL. Thirdly, the PA-RL’s performance is worse than PA-DRL algorithm, because reinforcement learning is worse than deep reinforcement learning when making accurate judgements about unknown situations in the sample. Last but not least, the variance of N2N algorithm is the highest, because the feedback functions of PA-DRL and PA-RL consider nodes’ load balancing by maximum CPU usage, these algorithms tend to deploy VNF on node with more resources. Thus, they are more advanced than N2N’s optimization function under each SFC requests number.

We use the D-value between maximum and average resource utilization to measure the network load balancing, therefore, the smaller the D-value is, the better the load balancing is. In order to measure the load balancing of nodes, Figure 7 shows the D-value between maximum and average node resource utilization of proposed PA-DRL algorithm and other four baseline algorithms. From Figure 7, it is easily to observe three facts. First of all, it can be seen that the performance of PA-DRL is the best, because PA-DRL utilizes deep reinforcement learning to deploy VNFs and backups, which is more efficient in optimizing nodes’ load balancing. Then, RND-BACKUP, ALL-BAKCUP, and PA-RL algorithms’ performances are similar, because PA-RL determines the backup scheme by priority, which is more advanced than RND-BACKUP and ALL-BACKUP, however, the optimization efficiency of reinforcement learning in large-scale unknown cases is weaker than that of deep reinforcement learning. Thirdly, the performance of N2N is the worst, because N2N is based on LP algorithm, which cannot optimize the target with better and better effect.

#### 4.3.6. Load Balancing of Network Links

Figure 8 shows the network link load variance for PA-DRL, ALL-BACKUP, RND-BACKUP, RL, and N2N models in terms of SFCs number. From Figure 8, it is easily to observe that the PA-DRL’s variance is the lowest like Figure 6, because we use maximum bandwidth usage as optimization goal in feedback function, the performance will be the best through learning process. The RND-BACKUP and ALL-BACKUP models’ curves are higher than PA-DRL, because the deployment location determining schemes of them are not better than PA-DRL, so the link which is chose to deploy virtual link is not the best, which leads to poor results than PA-DRL. The PA-RL’s curve is higher than PA-DRL, because deep reinforcement learning is better than reinforcement learning when making accurate judgements about unknown situations in the sample. The N2N’s performance is the worst, because the feedback functions of PA-DRL and PA-RL are considered links’ load balancing by maximum bandwidth usage, which will get better optimization result in large-scale SFC requests, so they are more advanced than N2N’s optimization function.

As mentioned above, in order to measure the load balancing of links, Figure 9 shows the D-value between maximum and average link resource utilization of proposed PA-DRL algorithm and other four baseline algorithms. Intuitively, there are three facts in Figure 9. Firstly, the D-value of PA-DRL is the lowest, because PA-DRL uses deep reinforcement learning to process deployment, which can optimize the target better compared with other algorithms, so the performance is the best. Secondly, RND-BACKUP, ALL-BAKCUP, and PA-RL algorithms’ performances are similar and better than N2N, because they all use machine learning to optimize target, which can get better performance in large-scale requests environment than LP algorithm. Finally, the performance of N2N is the worst, because N2N is based on LP algorithm, its optimization performance can only be maintained to a certain extent, and cannot get better results with the increase of SFC requests.

## 5. Conclusions

In this paper, we propose a novel PA-DRL for reliable SFC deployment. PA-DRL determines the backup scheme through the priority of SFC and network node, then PA-DRL use deep reinforcement learning algorithm to choose deployment and backup locations of VNF, and update the neural network dynamically. The results show that compared with the other four models (ALL-BACKUP, RND-BACKUP, PA-RL, and N2N), PA-DRL can improve the reliability of SFC and the efficient utilization of CPU resource. In addition, PA-DRL uses delay and network load balancing as feedback factors, so that it can reduce the transmission delay, and improve the load balancing of nodes and links to a certain extent.

Although our proposed PA-DRL method can achieve reliable SFC deployment, there are still some shortcomings that need to be resolved. At first, the PA-DRL deploys SFCs on the network topology which is designed in our work, the experiment results show that the performance of PA-DRL is better than others, however the network topology is too small to have limitations. Therefore, the PA-DRL model need to apply on another expanded network topology. Secondly, the parameters of neural network and deep reinforcement learning are designed through minor adjustments, including hidden layers, neurons number of neural networks, and weight coefficients of deep reinforcement learning, we can’t guarantee the performance is the best. Therefore, we need to adjust the parameters through more experimental comparison. Finally, we will apply PA-DRL method on actual network environment, deploy our algorithm on hardware devices to achieve the experimental results in line with actual conditions.

## Figures and Tables

**Figure 1 sensors-21-02733-f001:**
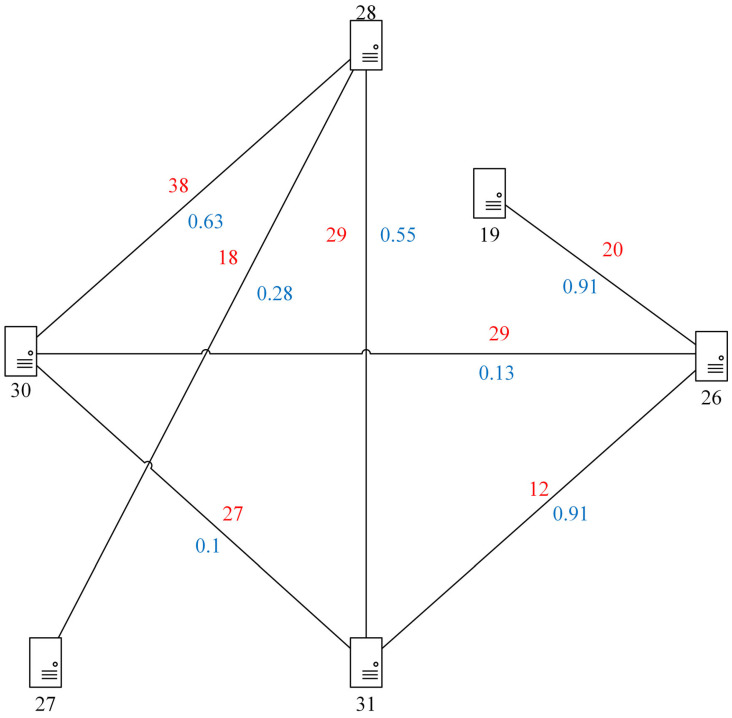
Network topology with CPU capacity, bandwidth capacity, and link delay.

**Figure 2 sensors-21-02733-f002:**
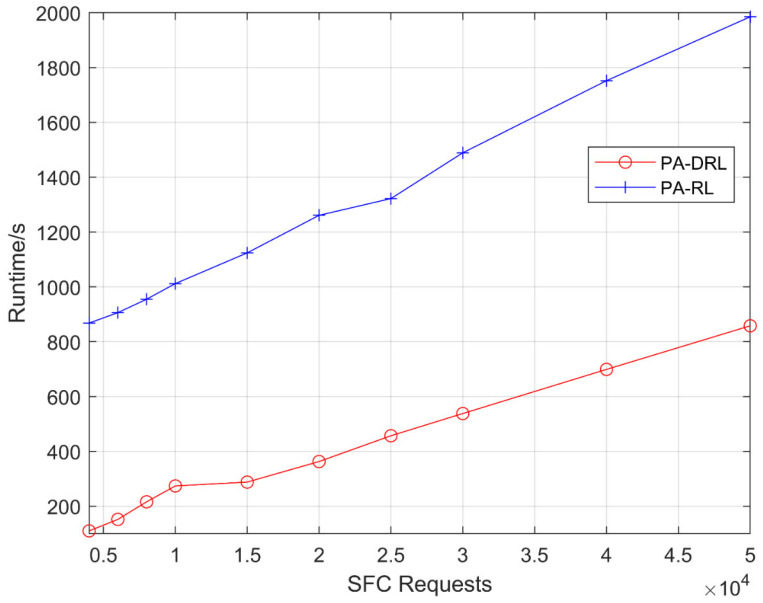
Runtime of PA-DRL and PA-RL under each SFC requests number.

**Figure 3 sensors-21-02733-f003:**
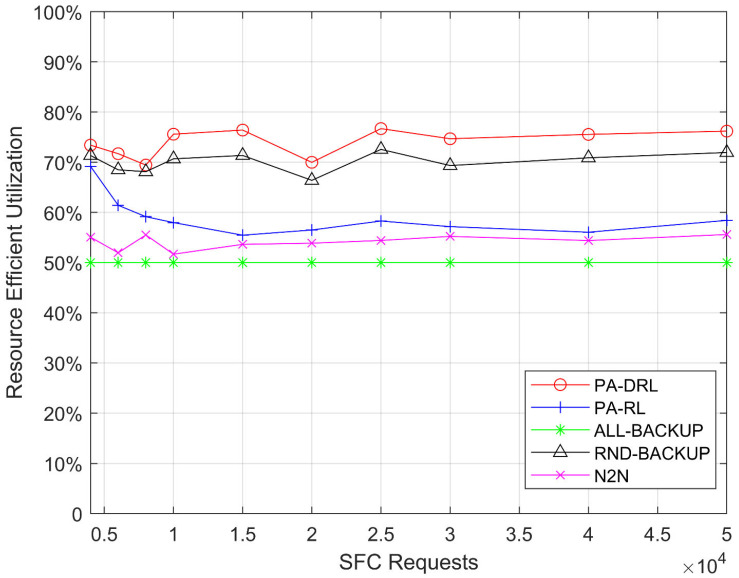
Efficient utilization of CPU resource under each SFC requests number.

**Figure 4 sensors-21-02733-f004:**
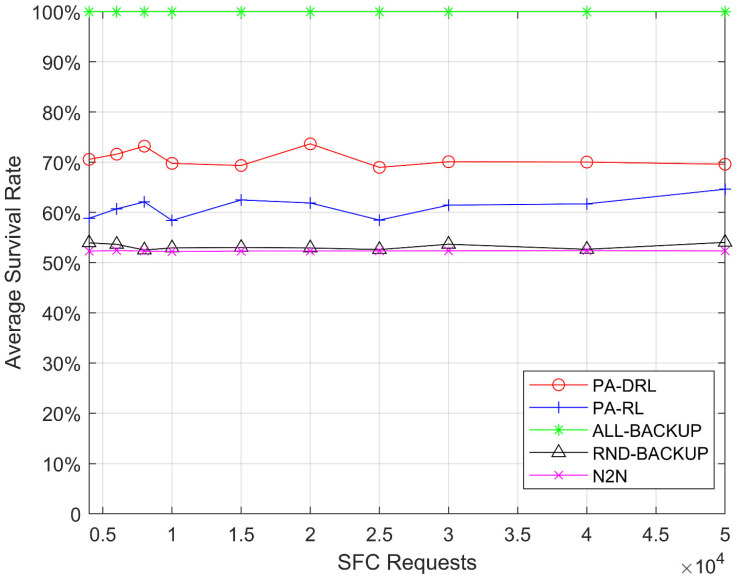
Average survival rate after failure of any node under each SFC requests number.

**Figure 5 sensors-21-02733-f005:**
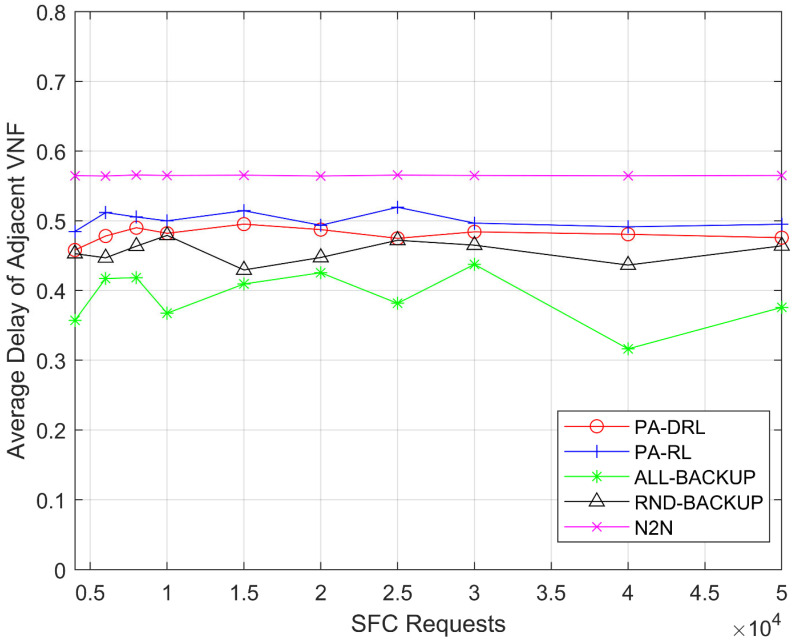
Average delay of adjacent VNFs under each SFC requests number.

**Figure 6 sensors-21-02733-f006:**
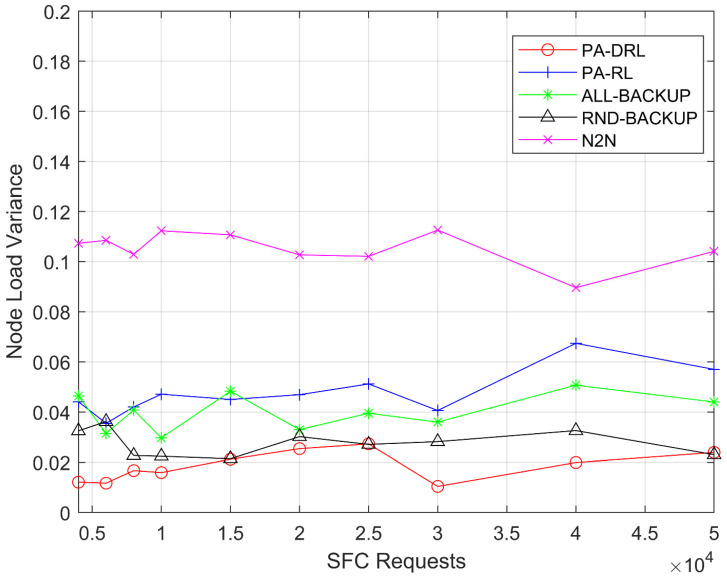
Node load balancing variance under each SFC requests number.

**Figure 7 sensors-21-02733-f007:**
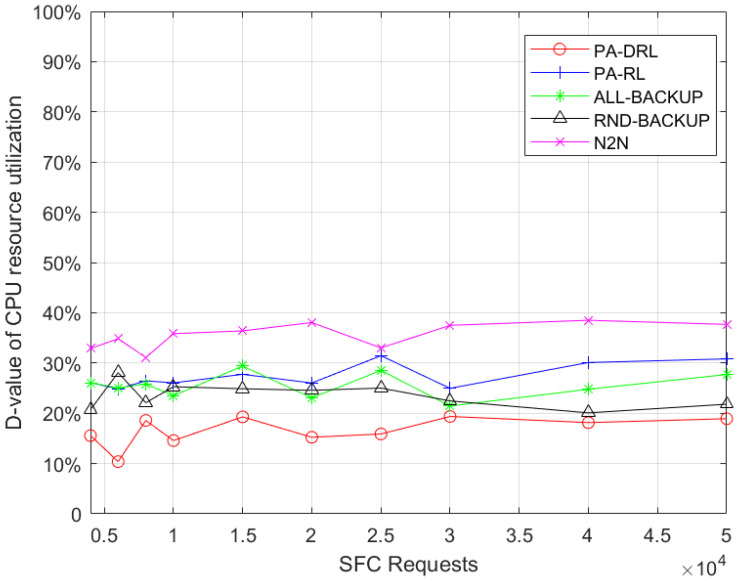
D-value between maximum and average nodes’ resource utilization under each SFC requests number.

**Figure 8 sensors-21-02733-f008:**
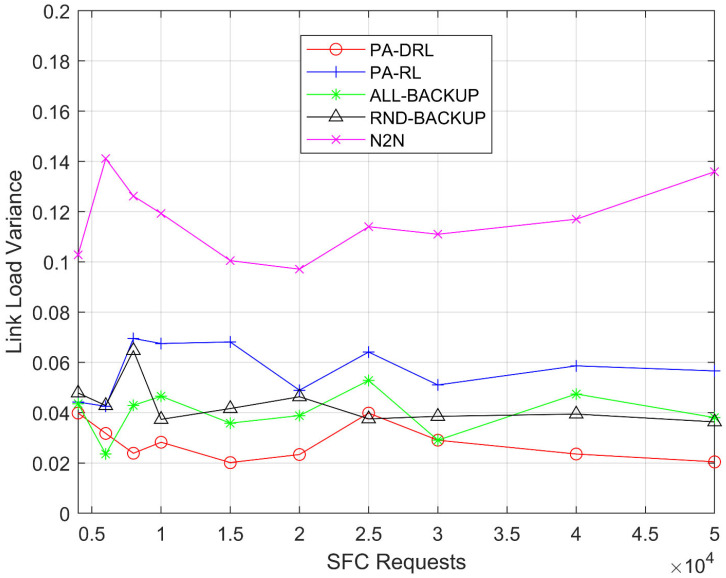
Link load balancing variance under each SFC requests number.

**Figure 9 sensors-21-02733-f009:**
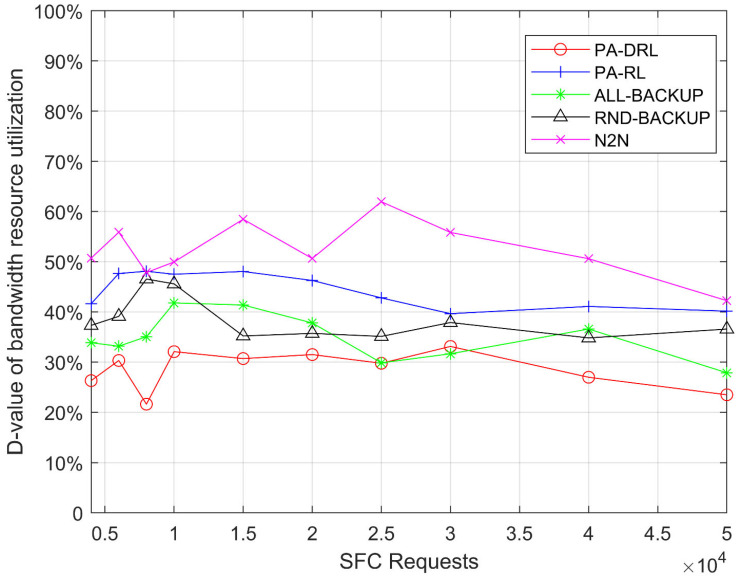
D-value between maximum and average links’ resource utilization under each SFC requests number.

**Table 1 sensors-21-02733-t001:** Backup scheme in different priority situations.

Priority of SFC and Node	Backup Scheme
$P_{S}=high,\;P_{N}=high$	$V_{b}\neq 0$
$P_{S}=high,P_{N}=low$	$V_{b}\neq 0$
$P_{S}=medium,P_{N}=high$	$V_{b}\neq 0$
$P_{S}=medium,P_{N}=low$	$V_{b}=0$
$P_{S}=low,P_{N}=high$	$V_{b}=0$
$P_{S}=low,P_{N}=low$	$V_{b}=0$

**Table 2 sensors-21-02733-t002:** All VNFs and their respective CPU resource requirements.

Varieties of VNF	CPU Resource Requirements
FW	1 unit
NAT	1 unit
DPI	2 units
DNS	3 units
LB	3 units

**Table 3 sensors-21-02733-t003:** All virtual links and their respective bandwidth resource requirements.

Varieties of Virtual Links between Adjacent VNFs	Bandwidth Resource Requirements
FW-NAT (NAT-FW)	1 unit
FW-DPI (DPI-FW)	1 unit
FW-DNS (DNS-FW)	2 units
FW-LB (LB-FW)	2 units
NAT-DPI (DPI-NAT)	2 units
NAT-DNS (DNS-NAT)	2 units
NAT-LB (LB-NAT)	2 units
DPI-DNS (DNS-DPI)	3 units
DPI-LB (LB-DPI)	2 units
DNS-LB (LB-DNS)	3 units

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
