# Peer review of "Reliable Service Function Chain Deployment Method Based on Deep Reinforcement Learning"

_sensors, 2021, doi:10.3390/s21082733_

Round 1
Reviewer 1 Report
The paper is about applying reinforcement learning for the SFC placement problem considering a backup plan.
I couldn't figure out how the authors model the routing process. Please explain where do you model the flow conservation constraints? (input and output to a node should be the same unless it is source or destination).
Please clarify why c_d and c_b may be different in eq (5)?
The logic behind the priority system (section 3.2) is ambitious to me. Based on this metric the algorithm decided whether to assign a backup node/link. Therefore, this metric is very important. However, in the current state, it is difficult to follow this sub-section. I would suggest, first have an overview of the logic behind this process (and briefly explaining how you would do that) then explain the technical details.
What is the meaning of "load variance" in section 4.3.6? The authors should investigate Max and Average link/node utilization of their solution.
It would be better if the authors have evaluated their solution under intensive load. In the current experiment, the avg load on nodes is under 12% (fig 6) and the avg on links is under 15% (fig 7).
The quality of figures is low and all plot show be replaced with a high-quality one.
Reviewer 2 Report
- The abstract must summarize the performance evaluation results.
- The related work is only descriptive (1 or 2 sentences per papers) and there are insufficient descriptions of the pros and cons of the work that is cited.
- Figures need to be amended, where the font size is slightly small to be seen which makes it difficult to read.
- The results should be further analyzed, more details and further discussion of the simulation results is needed.
Round 2
Reviewer 2 Report
The authors mention all my comments